# Trichilones A–E: New Limonoids from *Trichilia adolfi*

**DOI:** 10.3390/molecules26113070

**Published:** 2021-05-21

**Authors:** Mariela Gonzalez-Ramirez, Ivan Limachi, Sophie Manner, Juan C. Ticona, Efrain Salamanca, Alberto Gimenez, Olov Sterner

**Affiliations:** 1Department of Chemistry, Centre for Analysis and Synthesis, Lund University, 22100 Lund, Sweden; mariela_alejandra.gonzalez_ramirez@chem.lu.se (M.G.-R.); ivan.limachi_valdez@chem.lu.se (I.L.); sophie.manner@chem.lu.se (S.M.); 2Instituto de Investigaciones Farmaco Bioquimicas, Universidad Mayor de San Andres, La Paz, Bolivia; biojuancarlos_11@yahoo.com (J.C.T.); efrain_salamanca@hotmail.com (E.S.); agimenez@megalink.com (A.G.)

**Keywords:** *Trichilia adolfi*, limonoid, trichilones A-E, cytotoxicity, leishmanicidal activity

## Abstract

In addition to the trichilianones A–D recently reported from *Trichilia adolfi*, a continuing investigation of the chemical constituents of the ethanol extract of the bark of this medicinal plant yielded the five new limonoids **1**–**5**. They are characterized by having four fused rings and are new examples of prieurianin-type limonoids, having a ε-lactone which in **4** and **5** is α, β- unsaturated. The structures of the isolated metabolites were determined by high field NMR spectroscopy and HR mass spectrometry. The new metabolites were shown to have the ε-lactone fused with a tetrahydrofuran ring which is connected to an oxidized hexane ring joined with a cyclo-pentanone having a 3-furanyl substituent. As the crude extract possesses antileishmanial activity, the compounds were assayed for cytotoxic and antiparasitic activities in vitro in murine macrophage cells (raw 264.7 cells) and in *Leishmania amazoniensis* as well as *L. braziliensis* promastigotes. Metabolites **1**–**3** and **5** showed moderate cytotoxicity (between 30–94 µg/mL) but are not responsible for the antileishmanial effect of the extract.

## 1. Introduction

The Meliaceae family of plants includes species dispersed in tropical areas of America, Africa and Asia [1,2]. The *Trichilia* genus, part of this family, comprises several species that have been studied as they are used in traditional medicine, and phytochemical investigations have revealed that limonoids are typical constituents. The limonoids are tetranortriterpenoids, based on the triterpene class, which have been reported from several species of not only the Meliaceae but also the Cucurbitaceae and Rutaceae families. They are present, for example, in sweet- or sour-scented citrus fruit. Limonoids are characterized for having various carbon skeleta, being highly oxygenated and often substituted with a 3-furanyl moiety [3]. From *Trichilia*, different types of limonoids have been reported including azadirachtin and the trichilins (trichilin type, e.g., trichilin A from *T. roka*) [4,5], the trichisintons (phragmalines type, e.g., trichisinton A from *T. sinensis*) [6], the trichinenlides (mexicanolide type, e.g., trichinenlide A from *T. sinensis*) [7], trichavensin (prieurianin type, from *T. havannensis*) [8], hirtin (cedrelone type, from *T. americana*) [9] and protolimonoids (e.g., from *T. elegans*) [10]. A few structures are shown in Figure 1, and it is obvious that the limonoids can be structurally varied in an unlimited way. The compounds isolated in this investigation are shown in Figure 2.

*Trichilia* species are frequently used in traditional medicine. Bark, leaves, fruits and roots have been used to treat a broad variety of disorders, including stomach, diarrhea, dermatitis, taeniasis, pneumonia, lung, kidney and liver pains, as well to heal wounds [11]. The biological and pharmacological properties of *Trichilia* species have been reviewed by Longhini et al. [1], Garg [12], Curcino Vieira et al. [2] and Komane et al. [13]. More recently, limonoids have been studied for their effects on inflammatory disorders [14,15] and as cytotoxic agents [9,16,17]. Others report their activity in mouse lymphoma cells [18], where they inhibit proteins involved in oncogenesis and chemotherapy resistance [19]. Limonoids may therefore have a potential use as anticancer agents [20].

In a recent study by us, the four novel limonoids trichilianone A–D were isolated from the bark of *Trichilia adolfi* [21]. The structures of the trichilianones appear related to the hortiolide-type limonoids, with five fused rings. An ε-lactone is attached to a tetrahydrofuran ring that is connected to a bicyclo [5.1.0] hexane system, joined with a cyclopentanone containing a 3-furanyl substituent. Continuing our investigation of the constituents of *T. adolfi*, with the aim to characterize its antiparasitic metabolites, the present study is focused on the characterization of new metabolites present in the bark. The structures of the isolated metabolites were elucidated by spectroscopic analysis; HR-MS, and 1D and 2D NMR experiments (COSY, HMBC and NOESY correlations are summarized in Figure 3, Figure 4 and Figure 5). The cytotoxicity of pure metabolites was evaluated in in vitro cell cultures of murine macrophages (Raw 264.7 cells), while the leishmanicidal properties were evaluated in the *Leishmania amazoniensis* and *L. braziliensis* promastigotes.

## 2. Results and Discussion

### 2.1. Structure Determination of Trichilone A (***1***)

Trichilone A, (**1**, name proposed by us) was isolated as a white amorphous powder. The elemental composition of **1** was determined by LC-HRMS experiments, the [M + H]^+^ ion was observed at m/z = 617.2963 which corresponds to C_33_H_45_O_11_ (calculated 617.2962). The elemental composition C_33_H_44_O_11_ reveals that the unsaturation index of **1** is 12. The ^1^H and ^13^C 1D NMR spectra display signals for exactly 44 protons and 33 carbons, and according to their chemical shifts the structure of 1 should contain 5 carbonyls and two C-C double bonds suggesting that **1** has five rings. In addition, the 1D NMR data indicate the presence of one methoxy group, five oxygenated but saturated carbons, four non-protonated but saturated carbons, eight saturated CH-groups (of which three are oxygenated), three saturated methylene groups and nine methyl groups. The atom numbering used in this study (see Figure 2) follows the same system as in our previous work [21], which is based on an earlier study of hortiolide-type limonoids isolated from *Hortia colombiana* [22].

A careful analysis of the HMBC data gives critical information about the positions of the methyl groups as well as the core structure of **1**. The two methyl groups 24-H_3_ and 25-H_3_ are both singlets and both give strong HMBC correlations to C-4 and C-5, and to each other’s carbons. 24-H_3_ and 25-H_3_ are consequently geminal and attached to the non-protonated C-4, which according to its chemical shift is oxygenated but also connected to C-5. The singlet 19-H_3_ gives four HMBC correlations, to C-1, C-5, C-9 and C-10, and as C-10 is the only non-protonated carbon of these four, the 19-H_3_ methyl group is attached to this carbon. This reveals the bond between C-5 and C-10. The singlet 26-H_3_ gives only three HMBC correlations, to C-8, C-9 and C-14. The non-protonated C-8 displays a chemical shift that suggests it to be oxygenated, and it is consequently to this carbon that the 26-H_3_ methyl group is attached. Finally, the methyl 18-H_3_, which is also a singlet, gives HMBC correlations to C-12, C-13, C-14 and C-17, suggesting that is attached to the non-protonated C-13. This reveals the following partial carbon skeleton: C-24/25–C-4–C-5–C-10–C-9–C-8–C-14–C-13–C-18. Strong ^1^H-^1^H couplings between 9-H and 11-H (8.0 Hz) and between 11-H and 12-H (11.6 Hz), along with HMBC correlations from 9-H to C-8, C-10 and C-11, from 11-H to C-9 and C-12, and from 12-H to C-11 and C-13, closes the first ring. In this, the chemical shifts for 11-H/C-11 and 12-H/C-12 suggest that these two carbons are oxygenated, more specifically acylated. This is confirmed by the HMBC correlations from 11-H, 2″-H, 3″-H_3_ and 4″-H_3_ to C-1″, and internal COSY and HMBC correlations between 2″-H, 3″-H_3_ and 4″-H_3_ and their corresponding carbons demonstrate the presence of an isobutyric acid ester at C-11. 12-H. The methyl singlet 2′-H_3_ give HMBC correlations to C-1′, showing that C-12 has an acetoxy substituent. On the other side of C-13 we find C-17 (with one proton) as discussed above, which is connected to C-16 (with two protons) according to the ^1^H-^1^H couplings between 17-H and 16-Ha (8.7 Hz) and between 17-H and 16-Hb (11.1 Hz). This is confirmed by the HMBC correlations from 17-H to C-16, and between 16-H_2_ to C-17. 14-H and 16-H_2_ give HMBC correlations to the keto function, while 14-H in addition gives correlations to C-8, C-12, C-13, C-17 and C-18, which show the position and nature of C-15 and close the second ring. The missing substituent on C-17 is a 3-furanyl group, and this partial structure was determined in the same way as with the trichilianones [21] by COSY, HMBC and ^1^*J*(C,H) couplings constants. This forms the third ring accounting by itself for three unsaturations, and the NMR data are quite similar to those reported for other limonoids containing a 3-furanyl substituent [23]. COSY and HMBC correlations between 5-H and 6-H_2_, and these protons and C-6 and C-7, and the HMBC correlation of the methoxy protons to C-7 show that this carbon is not part of a lactone. We have already noted that C-1 is connected to C-10, and the COSY correlation between 1-H and 2-H_2_ in addition to the HMBC correlations between 1-H and C-3 and C-3 and between 2-H_2_ and C-1, C-3 and C-10 demonstrate this partial structure. Evidence that C-3 is part of an ε-lactone [24], the fourth ring, is shown by the weak but clear HMBC correlation between 24-H_3_ and C-3. All protons and carbons are now accounted for, remaining one oxygen and the formation of a fifth ring. As it has been determined that both C-1 and C-8 are oxygenated, the formation of a tetrahydrofuran ring with the last oxygen bond to these both carbons is the only way to finalize the structure elucidation. A similar ring system has been identified in limonoids from species belonging to the genus *Hortia* [25].

The relative stereostructure of **1** was determined by analyzing its NOESY 2D spectra. No NOESY correlation was observed between 18-H_3_ and 12-H, 14-H or 17-H, suggesting that the latter three protons are positioned on the opposite side compared to the methyl group (which would be α if the stereostructure in Figure 2 is absolute). This was confirmed by the correlations observed between 12-H and 17-H, and between both these protons and 14-H (all three β). However, 18-H_3_ gives clear correlations to 11-H, 16-Hb, 21-H, 22-H and 26-H_3_, showing that these are on the same side as C-18. 26-H_3_ correlates with 1-H and 9-H (all α), but not with 14-H, while 19-H_3_ correlate with 1-H, 2-Hb, 6-H_2_, 9-H and 25-H_3_. 25-H_3_ (α) correlates with 2-Hb, while 24-H_3_ (β) correlates with 5-H. Finally, the observed correlation between 1-H and 9-H (both α) establishes the suggested relative stereostructure as shown in Figure 2. The fact that 9-H, C-19 and C-26 are on the same side of the molecule, would bend the molecule along C-8/C-9 and explain the weak NOESY correlations observed between 12-H, 14-H, 5-H and 6-H_2_, even if they on paper appear to be too far apart. The corresponding configuration has also been suggested in hortiolide-type limonoids [25].

### 2.2. Structure Determination of Trichilone B (***2***)

Trichilone B, (**2**) was isolated as a white amorphous powder. **2** has the elemental composition C_34_H_46_O_11_, deduced by LC-HRMS experiments in which the molecular ion was observed at m/z = 631.3120 [M + H]^+^ (calculated for C_34_H_47_O_11_ 631.3118). Consequently, the unsaturation index of **2** is 12, as in **1**, but having an additional carbon and two hydrogens. NMR 1D ^1^H and ^13^C NMR data show many similarities with those of **1**, and it appears likely that **2** shares its major structure with **1**. A comparison of the NMR data show that **2** has an additional methylene group, and as **2** is missing the signals for the isobutyric acid ester of **1**, it appears that **2** has been replaced with a 2-methyl butyric acid ester. This was confirmed by the HMBC correlations of the surrounding carbons. Methyl 5″-H_3_, a doublet, correlate with C-1″, C-2″ and C-3″, while the protons of 4″-H_3_, a triplet, correlate with C-2″ and C-3″. The corresponding COSY correlations were also observed. 11-H gives a HMBC correlation to C-1″, placing this acyloxy group at C-11. The relative stereochemistry was determined by NOESY experiment. As with **1**, no NOESY correlations was observed between 18-H_3_ and 12-H, 14-H or 17-H, while correlations were observed between 12-H and 17-H, and also between both of these protons with 14-H. 18-H_3_ correlate with 11-H, 16-Hb, 21-H, 22-H and 26-H_3_, showing that these are on the same side of the molecule as C-18. 26-H_3_ also correlate with 1-H and 9-H, but not with 14-H, while 19-H_3_ correlate with 1-H, 2-Hb, 6-H_2_, 9-H and 25-H_3_. 25-H_3_ correlates with 2-Hb, while 24-H_3_ correlates with 5-H. The correlation observed between 1-H and 9-H establishes the suggested relative stereostructure as shown in Figure 2 and reveals that **2** has the same relative stereochemistry as **1**.

### 2.3. Structure Determination of Trichilone C (***3***)

Trichilone C, (**3**) was obtained as a solid amorphous powder. Its elemental composition was determined to be C_34_H_46_O_12_, by LC-HMRS experiments in which the molecular appeared at m/z = 647.3073 [M + H]^+^ (calculated for C_34_H_47_O_12_ 647.3068). Compared to **2**, **3** has one extra oxygen, and examination of the NMR 1D ^1^H and ^13^C NMR data suggests that a saturated methine on **2** has been hydroxylated in **3**. The signals for 45 protons can be observed in the proton spectrum, supporting the suggestion that a hydroxyl group with an exchangeable proton has been introduced. Seven of the 12 unsaturations are accounted for five carbonyl groups and two C-C double bonds, so **3** has consequently five rings just as **1** and **2**.

Most of the COSY and HMBC correlations discussed above for **1** and **2** can also be observed in the 2D spectra of **3**, suggesting that **3** has a similar structure. A major difference is the transformation of the 11-H from a double doublet to a doublet in the proton spectrum. 11-H only couples with 12-H in **3**, not to 9-H, and it appears as if C-9 is the carbon that has been hydroxylated, as a HMQC experiment revealed that C-9 is no longer protonated. By examining the correlations in the HMBC spectrum, it is evident that the protons in the vicinity of C-9 (1-H, 11-H, 19-H_3_ and 26-H_3_) correlate with a new carbon resonance at 89.2 ppm, which consequently is the hydroxylated C-9. When assigning the relative configuration of **3** using the data from NOESY experiments, the corresponding correlations as discussed above were noted. This includes correlations from 26-H_3_ to 1-H and 11-H, from 12-H to 14-H and 17-H, and from 19-H_3_ to 11-H, 6-H_2_ and 2-Hb. This suggest that the configuration of C-9 is the same as in **1**/**2**, and this is confirmed by the NOESY correlations from 12-H and 14-H to 5-H, caused by the folding of the molecule (also discussed above).

### 2.4. Structure Determination of Trichilone D (***4***)

Trichilone D, (**4**) was obtained as a white amorphous powder. The elemental composition of **4** is C_31_H_38_O_11_, which was determined by LC-HMRS experiments in which the molecular ion peak was observed at m/z = 587.2484 [M + H]^+^ (calculated for C_31_H_39_O_11_: 587.2492). The saturation index for **4** was consequently 13, one more than the previously discussed trichilones, and the examination of the NMR 1D ^1^H and ^13^C NMR spectra reveals the presence of five carbonyl groups and three C-C double bonds. This suggests that **4** consists of five rings. The ^13^C NMR spectra show a presence of ten methine carbons (including four olefinic and two oxygenated), two methylene groups, ten non-protonated carbons (including five carbonyl and three oxygenated carbons) and eight methyl groups (including a methoxy group). In **4**, C-9 is no longer hydroxylated as it demonstrates proton coupling between 9-H and 11-H. A major difference between **4**, and for example **1**, is the HMBC correlations observed from 19-H_3_, besides the expected correlations to C-5, C-9 and C-10, a correlation is also observed to an unsaturated carbon with the chemical shift 177.3 ppm. This should be C-1, and as also C-2 is unsaturated (with the carbon shift 95.1 ppm) in **4**, a main difference has been the introduction of a carbon–carbon double bond between C-1 and C-2 in the ε-lactone ring in **4**. The carbon chemical shifts of C-1 and C-2 of **4** are unusual, but this is caused by the double electronic effect of an electron withdrawing carbonyl group at C-3, making it α,β-unsaturated, and the oxygenation of C-1. Consequently, the ^1^*J*(C,H) coupling constants of C,H-2 is large (167 Hz). In addition, both C-11 and C-12 are acetoxylated as shown by the HMBC correlations from both 12-H and 2′-H_3_ to C-1′, and from both 11-H and 2″-H_3_ to C-1″. NOESY experiment determines that **4** conserve the same relative configuration as **1**.

### 2.5. Structure Determination of Trichilone E (***5***)

Trichilone E, (**5**) was isolated as a white amorphous powder. The elemental composition of 5 is C_33_H_42_O_11_, as deduced by LC-HMRS experiments in which the molecular ion peak was observed at m/z = 615.2798 [M + H]^+^ (calculated for C_33_H_43_O_11_; 615.2805) in positive-ion mode. **5** consequently contains 2 carbons and 4 hydrogens more than **4** but retains an unsaturation index of 13. As with **1** and **2**, the 1D NMR data of **4** and **5** are very similar (see Table 1 and Table 2). One of the acetyl groups of the former is obviously missing, and the signals for an isopropyl group have been added in the spectra of the latter. The HMBC correlations observed between 2″-H, 3″-H_3_, 4″-H_3_ and 11-H and C-1″ demonstrate that the acetyl group at C-11 in **4** has been exchanged for an isobutyric acid ester in **5**. NOESY experiments show that the relative configuration of **5** is the same as in **4**.

### 2.6. Biogenesis of the Trichilones

Not much is known about the biogenesis of this type of compounds, nor of the mechanisms of reactions involved in the biosynthesis of these highly complex molecules. Nevertheless, a suggestion for the biogenesis of **1–5** based on studies of similar compounds [26] is shown in Figure 6. The limonoids have a triterpene origin, and the intermediate **6** (an azadirone-type metabolite) can be imagined as the precursor (Figure 6). An oxidation of the C-7/C-8 bond to a lactone, followed by hydrolysis and a 180° bond rotation around C-9/C-10 would make a cyclization to form a tetrahydrofuran ring possible. A second oxidation of the C-3/C-4 bond could produce the ε-lactone ring and thereby the five-ring system. However, this should only be regarded as a possibility.

### 2.7. Leishmanicidal Activity and Cytotoxicity of the Trichilones

Table 3 summarizes the biological activities of the compounds isolated. The results shown that these compounds are not responsible for the activity observed for the crude extract (up to 50 μg/mL) on *Leishmania amazoniensis* and *L. braziliensis promastigotes* in vitro. A moderate cytotoxicity in murine macrophage cells was noted (IC_50_ values between 30 and 94 μg/mL), which is higher compared with the positive control used in the assay (miltefosine, 21 μg/mL).

## 3. Materials and Methods

### 3.1. General

Chemicals and solvents were bought from (Sigma-Aldrich, St Louis, MO, USA). 1D (^1^H, decoupled ^13^C) and 2D (COSY, NOESY, HMQC and HMBC) NMR experiments were recorded with a Bruker Advance spectrometer (Bruker Biospin AG, Industriestrasse 26, 8117 Fällanden, Switzerland) operating at 500 MHz for ^1^H and 125 MHz for ^13^C. All NMR experiments were carried out at 22 °C in CDCl_3_, and the solvent signals (at 7.27 and 77.0 ppm, respectively) were used as reference. IR spectra were carried out with an ALPHA FT-IR spectrometer (Bruker Biospin AG, In-dustriestrasse 26, 8117 Fällanden, Switzerland). Specific rotation in an ADP 450 series polarimeter apparatus (Bibby Scientific Ltd., Chelmsford, Essex, United Kingdom). LC-HRMS data were obtained with a Waters Aquity UPLC and Waters XEVO-G2 (CSH C_18_, 1.7 μm, 2.1 × 100 mm, Waters Corp, Milford, Worcester County, MA, USA) system. Silica gel 60 (20 mm × 300 mm, 70–230 mesh, Merck, Kenilworth, NJ, USA). Preparative HPLC separations were performed with an Agilent 1200 Infinity series system (Agilent, Santa Clara, CA, USA) equipped with a X-Terra prep RP-18 column (150 mm × 10 mm i.d., 5μm, Waters, Milford, MA, USA) at 25 °C and with the flow rate at 4.7 mL/min. The HPLC system was equipped with a diode array detector operating at 210 and 230 nm.

### 3.2. Plant Material

*Trichilia adolfi* Harms. (Meliaceae) was identified at the National Herbarium of Bolivia (LPB), where a voucher specimen (AS-092) is kept. It was collected in June 2017 (S 14°21′438″ W 67°34′728″) in the tropical humid forest in the North of La Paz, Bolivia.

### 3.3. Extraction and Isolation

Milled dried bark (1.0 kg) were macerated with 5.0 L ethanol for 72 h at room temperature to provide a crude extract after evaporation of the solvent (22.0 g, 2.2% of bark weight). This was then re-suspended in aqueous methanol (H_2_O: MeOH, 80:20, 0.5 L) and extracted three times with 0.5 L *n*-heptane, yielding 3.1 g (14% of the crude extract) after evaporation of the solvent. This was followed by the extraction of the aqueous methanol phase with CHCl_3_ (0.5 L), yielding 6.4 g (29% of the crude extract) after evaporation of the solvent, whereafter the aqueous phase was freeze-dried to yield 12.5 g (57% of the crude extract). The CHCl_3_ fraction was subjected to SiO_2_ gel column chromatography (1:30, w/w) eluted with mixtures of n-heptane: EtOAc (from 8:1 to 2:8 v/v). TLC analysis provided four main fractions from the least polar to the most polar: F1 (2.5 g), F2 (0.27 g), F3 (0.77 g) and F4 (0.68 g). Further fractionations were performed with fractions F1, F2 and F3 using a semi-preparative HPLC (*vide supra* for details) and eluted with mixtures of solvent A (3 mM formic acid in water) and solvent B (acetonitrile). The elution was performed in gradient mode, starting with A:B 85:15, reduced linearly to 77:23 in 5 min, a second linear decrease to 74:26 in 25 min, a third linear decrease to 60:40 in 30 min, and finishing with 54:46 for 45 min. All samples were filtered through 0.25 μm membrane filter prior to HPLC fractionations. From F2, crude Trichilone B (**2**) (5.1 mg, Rt = 15.35 min, 0.023% of raw extract) was obtained after SiO_2_ column chromatography eluting with n-heptane: EtOAc 8:2 (v/v, isocratic) and further purification by the HPLC procedure. From F3, Trichilone A (**1**) was isolated after SiO_2_ column chromatography of F3 eluting with n-heptane: EtOAc 6:4 (v/v, isocratic) to obtain a fraction that yielded pure 1 after HPLC purification (3.0 mg, Rt = 12.3 min, 0.013% of the raw extract). In the same fraction Trichilone C (**3**) was purified after HPLC isolation (6.3 mg, Rt = 18.3 min, 0.028% of raw extract). Finally, F4 was subjected to HPLC purification afforded Trichilone D (**4**) (12.0 mg, Rt = 15.8 min, 0.054% of raw extract) and Trichilone E (**5**) (14.6 mg, Rt = 18.0 min, 0.066% of raw extract).

### 3.4. Compounds Isolated

#### 3.4.1. Trichilone A (**1**)

White amorphous powder: [α]^20^_D_ = + 16 (c = 0.005, CHCl_3_). m.p. 143 °C. IR (film) νmax 2970 (C-H stretching), 1746, 1732 (carboxyl group) cm^−1^; ^1^H NMR (500 MHz, CDCl_3_) and ^13^C NMR (125 MHz, CDCl_3_) data, see Table 1 and Table 2, and Appendix A; LC-HRMS, 617.2963 [M + H]^+^ (calculated for C_33_H_45_O_11_, 617.2962)

#### 3.4.2. Trichilone B (**2**)

White amorphous powder: [α]^20^_D_ = + 18 (c = 0.005, CHCl_3_). m.p. 145 °C. IR (film) νmax 2956 (C-H stretching), 1734 (carboxyl group), 758 (C-H bending) cm^−1^; ^1^H NMR (500 MHz, CDCl_3_) and ^13^C NMR (125 MHz, CDCl_3_) data, see Table 1 and Table 2, and Appendix A; LC-HRMS, 631.3120 [M + H]^+^ (calculated for C_34_H_47_O_11_, 631.3118)

#### 3.4.3. Trichilone C (**3**)

White amorphous powder: [α]^20^_D_ = −20. (c = 0.005, CHCl_3_) m.p. 151 °C. IR (film) νmax 2976 (C-H stretching), 1746, 1731 (carboxyl group), 755 (C-H bending) cm^−1^; ^1^H NMR (500 MHz, CDCl_3_) and ^13^C NMR (125 MHz, CDCl_3_) data, see Table 1 and Table 2, and Appendix A; LC-HRMS, 647.3073 [M + H]^+^ (calculated for C_34_H_47_O_12_, 647.3068)

#### 3.4.4. Trichilone D (**4**)

White amorphous powder: [α]^20^_D_ = −13 (c = 0.005, CHCl_3_). m.p. 143 °C. IR (film) νmax 2985 (C-H stretching), 1742 (carboxyl group), 1682 (α,β-unsaturated carbonyl) cm^−1^; ^1^H NMR (500 MHz, CDCl_3_) and ^13^C NMR (125 MHz, CDCl_3_) data, see Table 1 and Table 2, and Appendix A; LC-HRMS, 587.2484 [M + H]^+^ (calculated for C_31_H_39_O_11_, 587.2492)

#### 3.4.5. Trichilone E (**5**)

White amorphous powder: [α]^20^_D_ = −10 (c = 0.005, CHCl_3_). m.p. 145 °C. IR (film) νmax 3001 (C-H stretching), 1743 (carboxyl group), 1685 (α,β-unsaturated carbonyl), 755 (C-H bending) cm^−1^; ^1^H NMR (500 MHz, CDCl_3_) and ^13^C NMR (125 MHz, CDCl_3_) data, see Table 1 and Table 2, and Appendix A; LC-HRMS, 615.2798 [M + H]^+^ (calculated for C_33_H_43_O_11_, 615.2805)

### 3.5. In Vitro Leishmanicidal Activity

Leishmanicidal activity was assayed according to Williams [27], with some modi-fication. Promastigotes of *Leishmania*: *L. amazonensis*, Clone 1, NHOM-BR-76-LTB-012 (Lma, donated by the Paul Sabatier Université, France) and *L. braziliensis* M2904 C192 RJA (M2904, donated by Dr. Jorge Arévalo from Universidad Peruana Cayetano Heredia, Peru). The strains were cultured in Schneider’s insect medium (pH 6.2), supplemented with 10% FBS and incubated in 96-microwell plates at 26 °C. Briefly, promastigotes in logarithmic phase of growth, at concentration of 106 parasites/mL, were exposed to samples dissolved in DMSO (1%) at different concentrations (3.1–100 μg/mL). Miltefosine (3.1–100 μg/mL, IDPS, France) was used as a positive control drug. The microwell plates were incubated for 72 h at 26 °C, the optical density of each well was measured and the IC_50_ values calculated. All assays were performed in triplicate.

### 3.6. Cytotoxicity

The Raw 264.7 murine macrophage cell line was purchased from the American Type Culture Collection (ATCC-TIB71) (Manassas, VA, USA). The cells were maintained in DMEM-HG medium supplemented with 10% fetal bovine serum, 100 U/mL of penicillin and 100 μg/mL of streptomycin and sodium bicarbonate (2.2 g/L) in humidified atmosphere at 37 °C with 5% CO_2_. Samples were prepared as described above and added (in 100 μL DMSO) at different concentrations (6.2–200 μg/mL). Medium blank, control drugs and cell growth controls were included to evaluate cell viability. The plates were incubated for 72 h at 37 °C with 5% CO_2_. After incubation for the indicated time, the cells were washed, after which 10 μL of Resazurin reagent (2.0 mM) was added. They were further incubated at 37 °C for 3 h in a humidified incubator. The IC_50_ values were assessed using a fluorometric reader (540 nm excitation, 590 nm emission) and Gen5 software. All assays were performed in triplicate.

## 4. Conclusions

The isolation and structure elucidation of five new limonoids from *Trichilia adolfi* was reported, and to our disappointment they were not found to possess antileishmanial activity. Not much is known about the biogenesis of this type of these highly complex molecules. Nevertheless, a suggestion for the biogenesis of **1**–**5** is shown in Figure 6. The limonoids have a triterpene origin, and the intermediate **6** (an azadirone-type metabolite) can be imagined as a precursor (Figure 6). An oxidation of the C-7/C-8 bond to a lactone followed by hydrolysis and a 180° bond rotation around C-9/C-10, may facilitate the formation of a tetrahydrofuran ring. A second oxidation of C-3/C-4 would produce the ε-lactone ring and thereby the correct five-ring system. Manipulation of the two carbon–carbon double bonds (C-1/C-2 and C-9/C-11) could then yield all five metabolites isolated in this investigation.

## Figures and Tables

**Figure 1 molecules-26-03070-f001:**
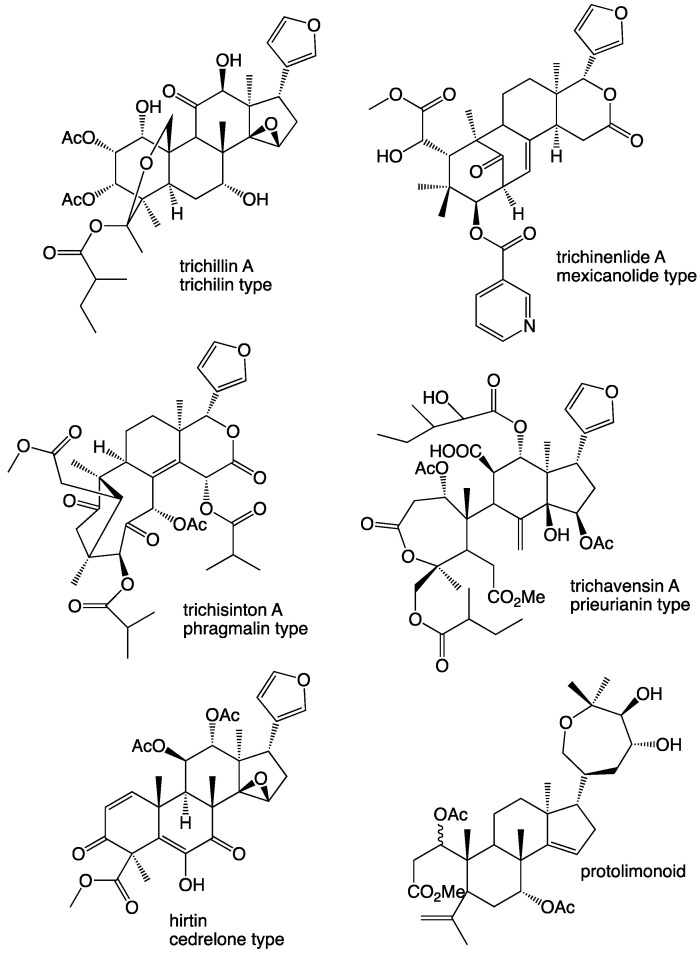
Limonoids isolated from other *Trichilia* species.

**Figure 2 molecules-26-03070-f002:**
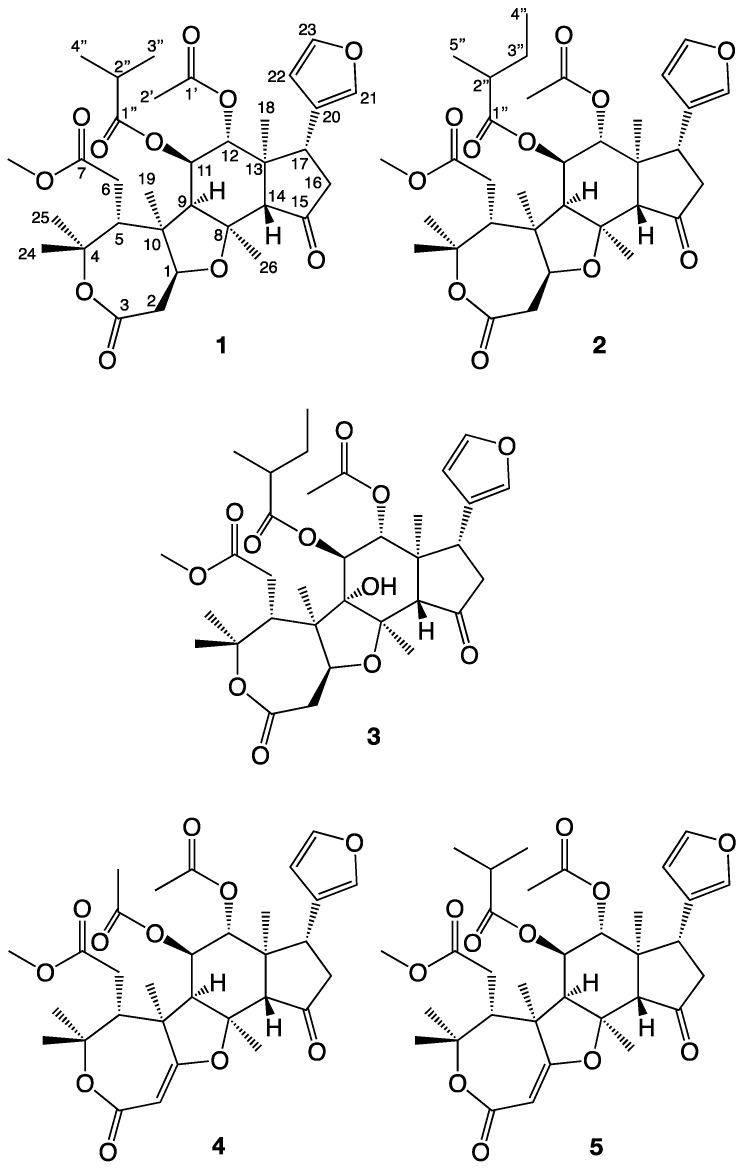
The new limonoids isolated from *Trichilia adolfi* in this investigation.

**Figure 3 molecules-26-03070-f003:**
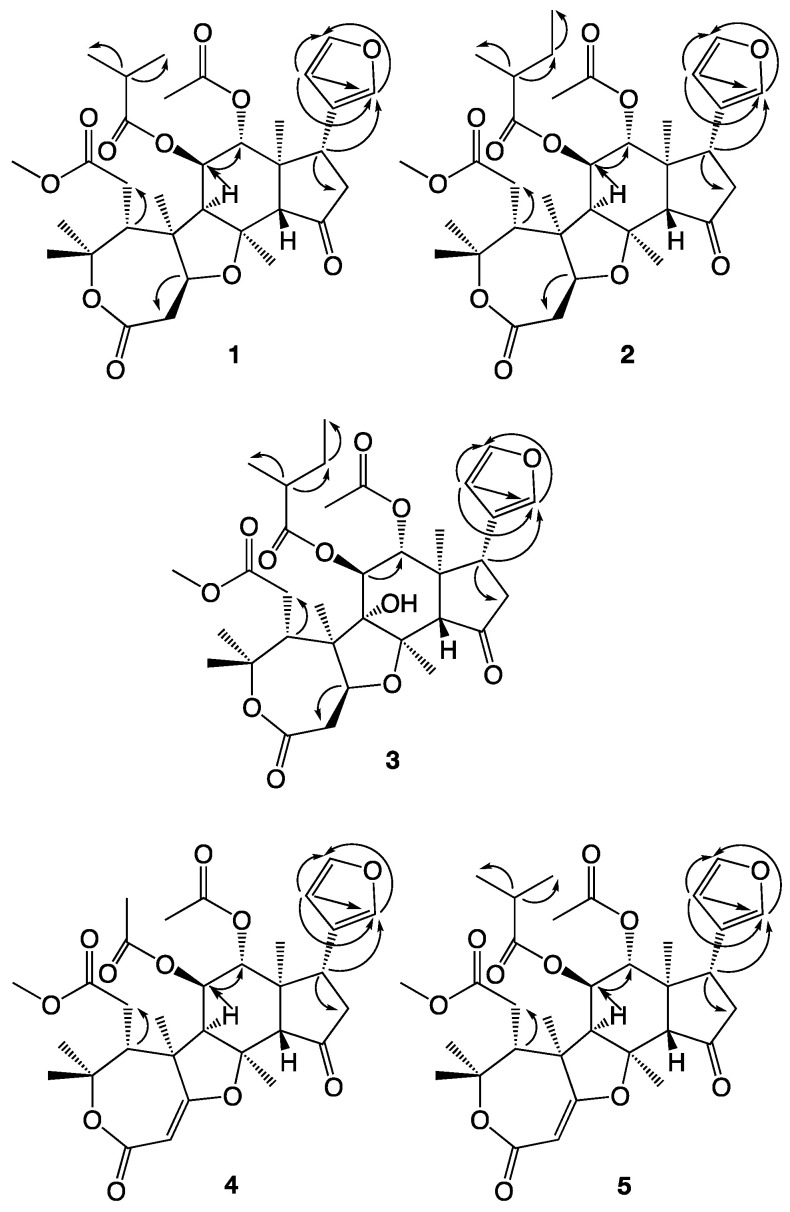
COSY correlations observed with compounds **1**–**5**.

**Figure 4 molecules-26-03070-f004:**
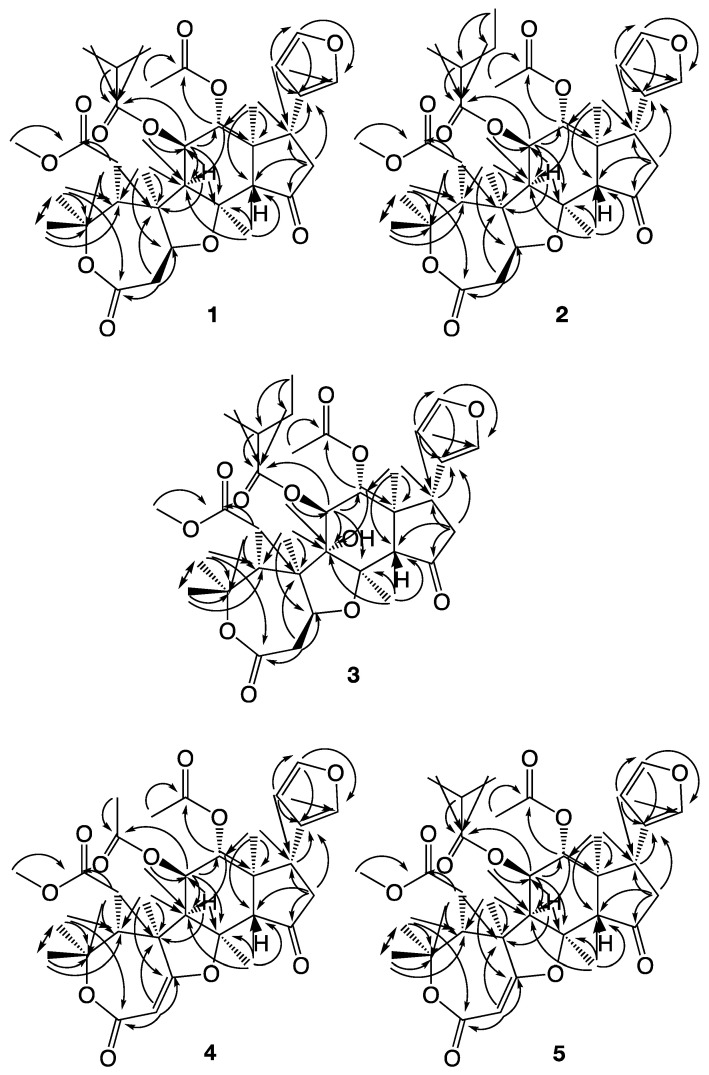
HMBC correlations observed with compounds **1**–**5**.

**Figure 5 molecules-26-03070-f005:**
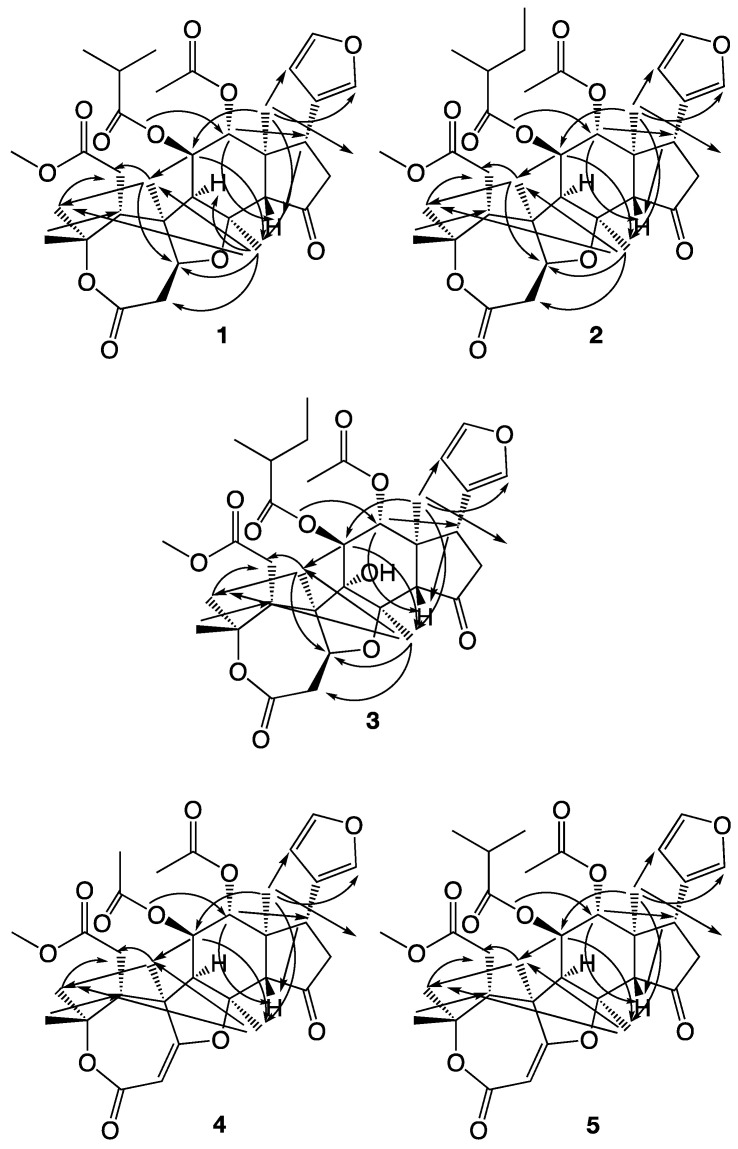
NOESY correlations observed with compounds **1**–**5**.

**Figure 6 molecules-26-03070-f006:**
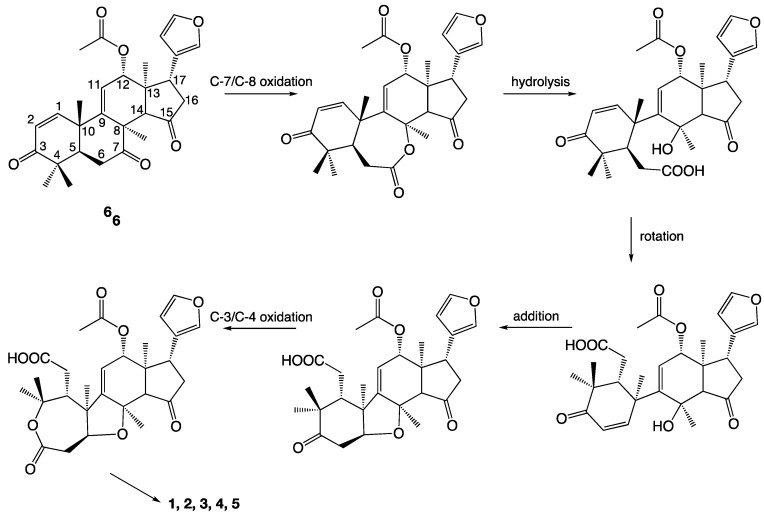
A possible biogenesis of compounds **1**–**5**.

**Table 1 molecules-26-03070-t001:** ^1^H NMR spectroscopic data for 1–5 recorded in CDCl_3_. The chemical shifts are in ppm, multiplicities and coupling constants (in Hz) are given in parentheses. The solvent signal at 7.27 ppm was used as reference.

H	1	2	3	4	5
1	3.63 (dd, 6.0, 2.2)	3.64 (dd, 6.0, 2.0)	4.18 (dd, 6.5, 2.1)	-	-
2a	3.17 (dd, 16.5, 6.0)	3.16 (dd, 16.5, 6.0)	3.15 (dd, 16.6, 6.4)	5.47 (s)	5.47 (s)
2b	3.10 (dd, 16.5, 2.2)	3.10 (dd, 16.5, 2.0)	3.04 (dd, 16.6, 2.1)	-	-
5	3.30 (dd, 10.5, 0.9)	3.30 (dd, 10.4, 0.8)	3.31 (dd, 10.4, 0.8)	3.72 (dd, 11.1, 0.7)	3.75 (dd, 11.3, 1.2)
6a	3.51 (dd, 18.1, 0.9)	3.50 (dd, 18.0, 0.8)	3.56 (dd, 18.4, 0.8)	2.89 (dd, 17.4, 11.2)	2.90 (dd, 17.4, 11.3)
6b	2.30 (dd, 18.1, 10.5)	2.30 (dd, 18.0, 10.4)	2.39 (dd, 18.4, 10.4)	2.34 (d, 17.4, 0.7)	2.34 (dd, 17.4, 1.2)
9	2.36 (d, 8.0)	2.36 (d, 8.0)	-	2.57 (d, 5.8)	2.59 (d, 5.6)
11	5.36 (dd, 11.6, 8.0)	5.36 (dd, 11.6, 8.0)	5.26 (dd, 11.2)	5.24 (dd, 12.0, 5.8)	5.24 (dd, 12.1, 5.6)
12	5.72 (d, 11.6)	5.72 (dd, 11.6)	5.80 (dd, 11.2)	5.64 (d, 12.0)	5.64 (d, 12.1)
14	3.17 (s)	3.17 (s)	3.24 (s)	2.37 (s)	2.38 (s)
16a	2.70 (dd, 19.1, 8.7)	2.70 (dd, 19.0, 8.5)	2.72 (dd, 19.1, 8.5)	2.74 (dd, 19.5, 8.9)	2.74 (dd, 19.5, 8.7)
16b	2.29 (dd, 19.1, 11.1)	2.29 (d, 19.0, 11.3)	2.35 (dd, 19.1, 7.0)	2.53 (dd, 19.5, 12.5)	2.52 (dd, 19.5, 12.3)
17	3.63 (dd, 11.1, 8.7)	3.63 (dd, 11.3, 8.5)	3.62 (dd, 8.5, 7.0)	3.59 (dd, 12.4, 8.9)	3.58 (dd, 12.3, 8.8)
18	1.02 (s)	1.02 (s)	1.06 (s)	1.04 (s)	1.06 (s)
19	1.12 (s)	1.13 (s)	1.12 (s)	1.63 (s)	1.63 (s)
21	7.23, (dd, 1.6, 0.8)	7.23, (dd, 1.6, 0.8)	7.24 (dd, 1.6, 0.8)	7.22 (dd, 1.6, 0.8)	7.21 (dd, 1.6, 0.9)
22	6.23 (dd, 1.8, 0.8)	6.23 (dd, 1.8, 0.8)	6.24 (dd, 1.8, 0.8)	6.22 (dd, 1.8, 0.8)	6.22 (dd, 1.8, 0.9)
23	7.37 (dd, 1.8, 1.6)	7.37 (dd, 1.8, 1.6)	7.37 (dd, 1.8, 1.6)	7.37 (dd, 1.8, 1.6)	7.37 (dd, 1.8, 1.6)
24	1.38 (s)	1.38 (s)	1.41 (s)	2.00 (s)	2.00 (s)
25	1.57 (s)	1.58 (s)	1.61 (s)	1.44 (s)	1.44 (s)
26	1.58 (s)	1.59 (s)	1.54 (s)	1.73 (s)	1.74 (s)
MeO	3.87 (s)	3.87 (s)	3.87 (s)	3.68 (s)	3.68 (s)
2′	1.62 (s)	1.61 (s)	1.58 (s)	1.85 (s)	1.81 (s)
2″	2.52 (dq, 7.0, 7.0)	2.34 (ddd, 7, 7, 7)	2.39 (ddd, 7, 7, 7)	1.90 (s)	2.26 (dq, 6.9, 6.9)
3″a	1.15 (d, 7.0)	1.66 (ddd, 14.6, 7.4, 7)	1.68 (ddd, 14.6, 7.4, 7)	-	1.09 (d, 6.9)
3″b	-	1.43 (ddd, 14.6, 7.4, 7)	1.47 (ddd, 14.6, 7.4, 7)	-	-
4″	1.14 (d, 7.0)	0.89 (t, 7.4, 7.4)	0.92 (t, 7.4, 7.4)	-	1.07 (d, 6.9)
5″	-	1.11 (d, 7.1)	1.11 (d, 7.0)	-	

**Table 2 molecules-26-03070-t002:** ^13^C NMR spectroscopic data for **1**–**5** recorded in CDCl_3_. The chemical shifts are in ppm, the multiplicities were determined by HMQC experiments. The solvent signal at 77.0 ppm was used as reference.

C	1	2	3	4	5
1	80.0; d	80.0; d	77.7; d	177.1; s	177.2; s
2	35.5; t	35.5; t	35.6; t	95.2; d	95.1; d
3	169.6; s	169.6; s	170.2; s	168.2; s	168.2; s
4	84.6; s	84.6; s	84.3; s	84.4; s	84.5; s
5	43.1; d	43.1; d	43.4; d	45.7; d	45.7; d
6	35.0; t	35.0; t	34.7; t	35.4; t	35.4; t
7	174.7; s	174.7; s	174.6; s	172.4; s	172.4; s
8	77.1; s	77.1; s	81.8; s	89.4; s	89.5; s
9	65.1; d	65.2; d	89.2; s	59.6; d	59.9; d
10	52.7; s	52.7; s	57.6; s	52.3; s	52.2; s
11	71.6; d	71.4; d	79.5; d	67.9; d	67.7; d
12	73.9; d	73.9; d	73.5; d	72.2; d	72.1; d
13	45.2; s	45.2; s	45.2; s	47.8; s	48.0; s
14	65.3; d	65.3; d	66.3; d	62.2; d	62.3; d
15	207.5; s	207.5; s	206.6; s	213.3; s	213.3; s
16	41.0; t	41.0; t	41.5; t	42.6; t	42.7; t
17	42.0; d	42.0; d	41.5; d	42.5; d	42.4; d
18	11.5; q	11.6; q	11.9; q	18.6; q	18.6; q
19	20.7; q	20.8; q	15.3; q	29.7; q	29.8; q
20	122.3; s	122.3; s	122.1; s	121.6; s	121.6; s
21	140.4; d	140.4; d	140.5; d	140.2; d	140.2; d
22	110.7; d	110.7; d	110.6; d	110.0; d	110.0; d
23	142.7; d	142.7; d	142.8; d	143.1; d	143.1; d
24	32.1; q	32.1; q	32.5; q	30.3; q	30.3; q
25	23.6; q	23.7; q	24.3; q	29.0; q	29.0; q
26	23.4; q	23.4; q	18.6; q	28.2; q	28.2; q
OMe	52.7; q	52.6; q	52.6; q	51.8; q	51.8; q
1′	170.0; s	170.0; s	169.6; s	168.9; s	169.1; s
2′	20.6; q	20.6; q	20.6; q	20.2; q	20.4; q
1″	175.9; s	175.5; s	178.1; s	170.2; s	177.3; s
2″	34.2; d	41.3; d	41.2; d	20.3; q	33.2; d
3″	18.9; q	26.2; t	26.3; t	-	18.8; q
4″	18.6; q	11.7; q	11.6; q	-	19.6; q
5″	-	16.6; q	16.3; q	-	-

**Table 3 molecules-26-03070-t003:** Leishmanicidal activity against leishmanial promastigotes (L.a: *Leishmania amazonensis*; L.b.: *L. braziliensis*) and cytotoxicity in murine macrophage cells (Raw 264.7 cells), of compounds **1**–**5** and the positive control miltefosine. The data are given as IC_50_ values in μg/mL (see Experimental section for details).

Compound	*L.a*	*L.b*	Raw
**1**	73.6 ± 5,1	58.0 ± 18.0	42.0 ± 0.9
**2**	>100	70.1 ± 3.0	30.0 ± 10.0
**3**	>100	87.2 ± 13.0	60.0 ± 17.0
**4**	>100	98.5 ± 1.5	94.1 ± 4.2
**5**	92.0 ± 7.0	68.1 ± 6.0	41.5 ± 24.0
Miltefosine	5.0 ± 0.2	4.07 ± 0.5	21.0 ± 2.0

## Data Availability

All data are available in this publication and in the Appendix A.

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
