# Peer review of "Trichilones A–E: New Limonoids from Trichilia adolfi"

_molecules, 2021, doi:10.3390/molecules26113070_

Round 1

Reviewer 1 Report

Manuscript ID: molecules-1189833

The manuscript of Mariela Gonzalez-Ramirez , Ivan Limachi , Sophie Manner , Juan C. Ticona , Efrain Salamanca , Alberto Gimenez , Olov Sterner as Co-authors: “New limonoids from Trichilia adolfi” report five new limonoids extracted from medicinal plant Trichilia adolfi.

The subject is of interest and would be useful for researchers working in the fields. Work is a further development of research reported in Molecules 2021, 26(4), 1019; https://doi.org/10.3390/molecules26041019, where four novel limonoids from the same plant were extracted.

The research reported in this article can be published in the Molecules after major revision.

Not looking at the fact that publication: Molecules 2021, 26(4), 1019; https://doi.org/10.3390/molecules26041019 was published on 15 February 2021 it was not mentioned in the references. Nothing also was given in the introductory part about this work. Only in the introduction, it was mentioned that this was an addition to work on previously reported trichilianones A-D. It seems that the manuscript was assembled in a hurry.

The introduction shouldn’t contain only general information about the usefulness Trichilia plant extracts, but also more information on the structures of previously known compounds structurally related to Limonoids. Please use as an example the previous article of your group about Trichilianones.

  1. The manuscript has an enormous amount of typing errors: put numbers of the compounds should be bolded in the text, use subscript for numerals when write CDCl3 or for any other summary formulae, use superscript for numerals in words like: 1D 1H and 13C NMR. Authors should also correct unnecessary word-breaks in all the text of the manuscript. It seems that the text of the manuscript was not correctly transferred into the template.
  2. Not looking at all above mentioned experimental part was performed accurately with necessary spectral data placed in supplementary materials. The manuscript contains also correct conclusions.

Based on the considerations detailed above, my recommendation is to reconsider this manuscript after major revision.

Author Response

Sirs,

I hereby resubmit the manuscript molecules-1189833:

"Trichilones A - E: New limonoids from Trichilia adolfi"

by Mariela Gonzalez-Ramirez, Ivan Limachi, Sophie Manner, Juan C. Ticona, Efrain Salamanca, Alberto Gimenez and Olov Sterner,

for reconsideration for publication in Molecules. I am sorry that it has taken so long for me to do this revision, but I have been severely ill.

Lund May 13th, 2021

Olov Sterner

Responses to the reviewers comments:

Reviewer 1:

  1. Our previous publication was mentioned among the references, reference 21. The reason for splitting the work into two publications was that the compounds have different carbon skeleta, and that that the structure elucidation of these structures is not trivial, especially in the amounts that are available. The reasoning of the spectroscopic details must be carried out very carefully, and that takes up space.

However, I have more clearly pointed out that a related publication exists.

  1. I have inserted a new Figure 1, in which examples of various limonoids obtained from Trichilia species are given.

  1. I have gone through the manuscript carefully to correct all errors like this. However, I am not able to control the powerful autoformatting capabilities of the manuscript template. This manuscript was prepared with a Macintosh computer, and if I save it and attach a copy with an e-mail to myself in order to continue working at home, the transfer introduces so many changes that the file is useless. In order to give you an impression what the resubmitted manuscript looked like, I have included a pdf version that should be correct.

Reviewer 2:

  1. I agree fully, but this was caused by the autoformatting of the template. There is a built-in function in the template for this, but for some reason this has gone bananas in this case. I am sorry about this, and hope that things will work better this time.

Reviewer 3:

  1. I have gone through the manuscript carefully in order to correct this, both the use of italics, bold, sub- and superscript. I can assure you that these errors were not present in the originally submitted version, they were introduced during the processing.

  1. I have gone through the manuscript carefully, and polished the English. I have deleted some parts of the discussion about the structure elucidation, and referred to our previous publication.

  1. A new title is proposed.

  1. New compounds have been given names.

  1. "Reports about the use of" has been changed. Most "as well as" have been omitted.

  1. This sentence has been changed.

  1. Se above for the hyphenation problem.

  1. This sentence has been omitted.

  1. Well, it is my experience that one has to be very systematic and careful when elucidating structures of this kind. If you want to prove a structure, you need to give all the detail, not too much rely on similarities with known compounds. I have therefore not changed this sentence.

  1. The spectroscopic data of some partial structures can readily be compared, and the best example is the 3-furanyl group. This is noted.

  1. That has been done.

  1. No attempts were made to solve the absolute configuration, although it should not be impossible. However, we needed the small amounts available to assay the biological activities.

  1. No, not really, This is very poorly investigated, and I can understand why. The discussions published about the biogenenis of the limonoids are as speculative as the one presented here. I agree about the Michael addition, and I have changed this to simply "addition".

  1. The summaries of the 2D NMR correlations are exactly as was required for the first publication (ref 21), and I feel that it would be correct to follow that format also here.

  1. The format of Table 3 has been corrected.

  1. Higher. As we used mg/ml in the first publication, we want to facilitate comparison by using it also here.

  1. I can not find "um" in the text.

18, No DEPT experiment with compound 1 was ever recorded, as the HMQC data was considered sufficient. The comment about a DEPT experiment with compound 1 has been omitted.

  1. Nonprotonated carbons will not give a signal in the DEPT spectrum, so I would assume that it is noise. However, C-20 of 2 has the chemical shift 122.3.

Reviewer 2 Report

The present manuscript reports the isolation and characterization of new limonoids natural products. The chemical structures of the compounds are adequately assigned based on the detailed spectroscopic analyses. The data are sufficiently provided. The proposed biosynthetic pathway for generating the unique skeleton is reasonable. This reviewer recommends acceptance of this manuscript for publication after addressing some necessary corrections and modifications.

The specific comment is indicated below.

1) Hyphenations of the words should be deleted unless necessary. Other typographical errors such as unnecessary space should be corrected. Please check them in the text thoroughly.

Author Response

(The authors gave the same response as above.)

Reviewer 3 Report

This manuscript describes the isolation and structure elucidation of five new limonoids (1-5). First, the manuscript contains format error. The plant name should be Italic but the authors only use Italic format in the title, not the whole text, such as P2L48. In addition, in vivo should be set in italic. For some reason there should not be a “-” in the word, such as P1L29, constitu-ents. In this whole paper, this situation happened in every page. In addition, the compound number should be in bold. The chemical formula should be in the correct editing, such as L2P62. Except the above mentioned, there are still other text and format errors. Some of the English is rather awkward in places. Hereby, I suggest the author should read the manuscript again carefully and re-write some sentences.  In addition, the authors need to detele some texts for the eculidation structure part as part of the structure is already published recently. 

I also encourage the authors to address the comments below.

The title should be more descriptive for this manuscript.

You should give a compound name for the new compounds 1-5, such as trichilianones E

P1L38 The English of “Reports about the use of “ is awkward. Too many times to use “as well as””

P1L50 Revise “The structures of the metabolites isolated were elucidated after an extensive spectroscopic analysis, high-resolution mass spectrometry as well as 1D and 2D NMR experiments were used as detailed below to elucidate these rather complex structures.” that sentence is very awkward and should be rewritten. And use HR-MS not high-resolution mass spectrometry.

P2L55 Delete (see Ex-perimental section for de-tails)

P3L65, delete car-bon-carbon.

L69, delete Further analysis of COSY, HMBC and NOESY NMR were used as support to elu-69 cidate the structure.

L73, concise the whole text for this paragraph as the structure are similar with hortiolide-type limonoids and the paper published recently from your group.

L119, can the authors provide some references and comparison with NMR data to confirm the structure?

P3L121, you should point the α-orientations or β for these protons to explain the relative configuration.

Have the authors tried to solve the absolute configuration? I think it would be great if you can try to explain some absolute configuration as the lack of structural novelty.

L294, can you provide any references to support the proposed biogenesis? And this is just speculation about the rearrangement of triterpene. I do not think step 4 is Michael addition. If the authors think so, the reference should be listed.

For the figures 3 and 4, too many correlations have shown in one structure. You only need draw selective important correlations.

Table 3, please use the correct format.

P11L311, lower or higher? Why do not use mM instead of μg/ml which It is easier to get a sense of either relevance, or future potential.

L362, change um to μm

Supplementary data, please add content in the first page so we can easily find the corresponding figure.

Please add the DEPT spectra for compound 1 to Supplementary information as you mentioned to use the DEPT data in P2L63.

In DEPT135 for compound 2, there in a negative peak around 120ppm. Is this signal C-20?

Author Response

(The authors gave the same response as above.)

Round 2

Reviewer 1 Report

Accept in present form.

Just one remark, supplementary materials were submitted as non-published material, please correct it.

Reviewer 3 Report

Just remember to double check the format for tables 1 and 2 (the lines are missing from my computer, you can follow the format as table 3)